# Brain Connectivity Analysis in Distinct Footwear Conditions during Infinity Walk Using fNIRS

**DOI:** 10.3390/s23094422

**Published:** 2023-04-30

**Authors:** Haroon Khan, Marco Antonio Pinto-Orellana, Peyman Mirtaheri

**Affiliations:** Department of Mechanical, Electronics and Chemical Engineering, OsloMet–Oslo Metropolitan University, 0167 Oslo, Norway; haroonkh@oslomet.no (H.K.);

**Keywords:** effective connectivity, footwear, functional near-infrared spectroscopy (fNIRS), Infinity Walk, pronation

## Abstract

Gait and balance are an intricate interplay between the brain, nervous system, sensory organs, and musculoskeletal system. They are greatly influenced by the type of footwear, walking patterns, and surface. This exploratory study examines the effects of the Infinity Walk, pronation, and footwear conditions on brain effective connectivity patterns. A continuous-wave functional near-infrared spectroscopy device collected data from five healthy participants. A highly computationally efficient connectivity model based on the Grange causal relationship between the channels was applied to data to find the effective relationship between inter- and intra-hemispheric brain connectivity. Brain regions of interest (ROI) were less connected during the barefoot condition than during other complex walks. Conversely, the highest interconnectedness between ROI was observed while wearing flat insoles and medially wedged sandals, which is a relatively difficult type of footwear to walk in. No statistically significant (*p*-value <0.05) effect on connectivity patterns was observed during the corrected pronated posture. The regions designated as motoric, sensorimotor, and temporal became increasingly connected with difficult walking patterns and footwear conditions. The Infinity Walk causes effective bidirectional connections between ROI across all conditions and both hemispheres. Due to its repetitive pattern, the Infinity Walk is a good test method, particularly for neuro-rehabilitation and motoric learning experiments.

## 1. Introduction

The first contact point of human feet with the ground is the heel bone, which is the beginning and crucial point for understanding human balance. As a result of improper heel bone placement, improper distribution of joint forces can lead to impairment of the musculoskeletal system, which can cause disabilities and high health costs at later stages [1]. Pronation describes how the foot normally rolls inward as we walk or run. It is part of the human body’s natural movement to absorb shocks and is adept at walking on uneven surfaces, and it differs from person to person [2]. If the foot rolls inward too far, it is called overpronation. This phenomenon is also known as the valgus of the heels and is a standard measure of the pronated condition. However, it is controversial to what degree the pronation of the foot affects the biomechanical chain and how to define the boundary between normal pronation and an overpronated rear foot angle [3,4,5]. From a neurological perspective, the sensory information collected from the foot, ears, and eyes is processed in several brain segments, including the midbrain and forebrain. However, the primary research on isolating pertinent brain areas involving specific functions is still debated [6]. The brain structure we usually refer to as singular has virtually bilateral symmetry on each side. More remarkably, each hemisphere in the forebrain is responsible for controlling motor responses for and sensory processing information from the opposite side of the body [7]. In the current study, we performed Infinity Walking with different footwear and medial heel wedge angles to examine the effect of different footwear and wedge angles (overpronation) on the brain connectivity between the hemispheres (both inter- and intra-hemispheric connectivity). We also aimed to observe the effect of the Infinity Walk on connectivity patterns. The term connectivity refers to Granger’s causal connectivity analysis.

The Infinity Walk, also referred to as the pattern eight walk, was introduced in the mid 1980s by clinical psychotherapist Deborah Sunbeck [8,9]. It is designed to increase motor, sensory, and cognitive abilities for a balanced walk and improve neuromuscular coordination [10]. The Infinity Walk was selected because it involves naturally occurring left and right directional shifts in foot placement, ankle joint rotation, and coordination between both sides of the body. The alternating shifts between the right and left sides of the body generate neuronal activation in both sides of the hemispheres [8,11]. The Infinity Walk challenges the balance on each side of the figure eight, while the sensorimotor system is actively challenged. As a result, the ankle joint and its stability play a more significant role in such walks. Postural stability and neural activation during walking are directly connected with ankle joint stability and heel bone (the calcaneus bone) placement [12,13]. In this study, we focused on the effect of a medial heel wedge to reduce overpronation and its effect on connectivity and cortical activation. A medially wedged insole can help reduce overpronation. Shoes with more medial wedge support (anti-pronated) than a neutral shoe produce forces resonating in the lower limb joints [14]. Thus, a medially wedged insole affects the biomechanics of the lower limb during walking [14], reducing ankle eversion angle and knee and hip motion in the transverse plane and consequently affecting the motion and movements that cause lower limb injuries [15]. Although some of the consequences have been studied in the literature, cortical activation and its network connectivities have limited reporting.

To examine the connection between overpronation and cortical activation, a robust and mobile brain imaging modality is required to monitor brain neuronal and hemodynamic changes during Infinity Walks. Modern imaging techniques have led to exciting developments in establishing various brain structures without separating the brain tissue. Among many brain imaging modalities, the most practical brain imaging modalities during human gait are electroencephalography (EEG) and functional near-infrared spectroscopy (fNIRS) [16,17,18]. Functional near-infrared spectroscopy (fNIRS) is a low-cost, easy-to-use, safe, portable, and non-invasive brain imaging modality for continuous measurement of hemodynamics in the cerebral cortex of the human brain [19]. The fNIRS research investigating walking and cortical activation is in its primary stages, such as simple walking vs. resting state classification [20]. In a cross-sectional fNIRS study investigating cortical activation in individuals with chronic ankle instability compared with the control group and single-leg stance position, significant differences in cortical activation variability were found in the supplementary motor area (SMA) compared to the control group [21]. The more significant SMA activation in individuals with chronic ankle instability demonstrates a potentially more significant change in strategy to maintain balance. Low cortical activation was observed during the barefooted walk on a hard surface compared to a complex walk on a challenging mat [22,23]. Variations in cortical activation are evident in many studies as alteration of neural mechanisms of postural control [13,21,22]. However, describing how different brain regions and hemispheres interact is still a matter of further discussion. The literature addressing how pronation and Infinity Walking affect brain connectivity is very limited. However, it is known from an EEG perspective that EEG rhythms and cortico–muscular relationships are connected during standing and walking [24]. Further, the study suggests that frontal–centroparietal regions are connected during walking. It has been previously reported that the sensorimotor region (C3 and CP3) is connected during standing in one position, with additional connectivity of centroparietal (Cpz) and frontal (Fz) areas during walking [24]. Premotor, cingulate, supplementary, and motor areas demonstrate a significant increase in ΔHbO in healthy people during treadmill walking. Still, the difference in walking patterns and pronation effects has not yet been explored [25]. Thanks to a Granger causal model, we can analyze effective connectivity and determine how cortical regions contribute to walking [26].

This work focused on examining the effective connectivity (Granger causality) between different brain regions during Infinity Walking with different footwear, particularly medially wedged support shoes to reduce overpronation during Infinity Walking. We examined the inter- and intra-hemispheric connectivity of Brodmann areas of interest with different footwear during Infinity Walking. First, it is important to understand how pronation affects our control strategy with different degrees of overpronation in the motor cortex. Second, different efforts, such as dual-tasking during walking, have been implemented to enhance cortical activation in many rehabilitation studies [27]. We hypothesize that complex walks such as Infinity Walks can improve neuroplasticity and can be used for rehabilitation. In this study, the barefooted walk was considered as a baseline or a comparison condition to other conditions.

## 2. Materials

### 2.1. Participant Selection and Ethical Consideration

Healthy participants (one male and four female) with no reported medical or psychological limitations were recruited for the experiment. The age range of the subjects was set to 20–30 years old, as there can be age-bound statistical differences in the pressure distribution and skin sensitivity between young adults and higher age groups [28]. The selection criteria for this study included participants with at least one slightly or heavily everted heel. In addition, it included light pronators, who are difficult to detect with the untrained eye. The Achilles tendon angle was determined using an Achilles marker and laser to facilitate the recruitment of volunteers and to assess the degree of pronation. The Achilles marker was a GAITLINE AS custom-made tool that marks the center-line of the Achilles tendon along the heel bone (calcaneus) while the subject is lying face down. This line matches the custom-made Achilles laser tool, which uses vertical lasers. The instrument is a frame placed around the subject’s foot that indicates the vertical axis along the leg in the frontal plane. The laser line is aligned with the traced mark by placing medial wedges of the degree required under the heel bone. Thus, the degree of correction needed is determined for both feet. The use of the Achilles laser tool is illustrated in Figure 1. Table 1 describes the degree of pronation calculated for each subject. The experimental protocol was approved by the REK (Regional Committee for Medical and Health Research Ethics, reference No. 322236) and the NSD (Norwegian Center for Research Data, reference No. 751430). The experiment was performed according to the latest Declaration of Helsinki.

### 2.2. Experimental Paradigm and Instructions

Before the experiment, participants were given explicit instructions about the experimental protocol, and demonstrations were conducted to familiarize them. Before fNIRS experimentation, the Achilles was marked, and the degree of pronation was determined as described in Section 2.1. A Qualisys motion capture system was used to track the lower extremity’s bio-mechanics with 32 markers; however, discussion of the motion tracking system is beyond the scope of this work. The experimental protocol consisted of initial and final rests for 17 s in the standing position, a single task block—defined as a task period of 22 s to complete one Infinity Walk starting at the central point of the pattern eight—and rests between task blocks of 10 s in the standing position. Since gait speed and variability affect the brain’s connectivity, we asked the participant to walk at an average speed to complete a round of the pattern eight in the task period [29]. The participants were asked to perform three trials before the start of the actual experiment to calculate the average time for completing one round of the Infinity Walk. The average speed of completing one round across the subjects was 18.9±1.54 s. The data for each condition mentioned in Table 2 were recorded separately. This is because it is difficult to change shoes during recording. The task performed represented the specific condition being tested. The experimental paradigm included four repetitions of trials, as illustrated in Figure 2. For example, in the barefooted condition, the barefooted run was repeated four times in one experimental protocol. The start/stop of the task was announced using a speaker. This experiment was designed so that participants were not distracted by the outside environment and did not have to exert extra effort.

### 2.3. Montage and Data Acquisition

The fNIRS optode location decider (fOLD v2.2) tool was used to arrange the optodes over the regions of interests (ROI), i.e., Brodmann areas 1–7 [30]. Optodes were placed over the motor and somatosensory cortex according to the 10-10 international system using the ICBM 152 head model [31] as shown in Figure 3. Further explanation of the brain regions is summarized in Table 3. A continuous-wave fNIRS device, NIRSport 2 (NIRx Medizintechnik GmbH, Berlin, Germany), was used to acquire the data with a sampling frequency of 10.2 Hz. The device uses two infrared light wavelengths (760 nm and 850 nm) with 32 optodes (sixteen sources and sixteen detectors). Short-channel data were also acquired during the experiment but were unreliable for noise removal. All signals were then filtered using an FIR(71) band-pass filter in the frequency range of 10–400 mHz.

## 3. Connectivity Model

This section describes the theoretical basis of our connectivity model. Assume a time series $Y\left(t\right)\in R^{M}$ with *L* channels and *T* time points. Under this model, each channel k=1,…L is associated with a 3D coordinate $c_{k}=\left(x_{k},y_{k},z_{k}\right)$. For a straightforward generalization, $c_{k}$ will be defined as coordinates in an ideal spherical geometric space. Our fNIRS data correspond to the mean position between the source and detector coordinates. The changes in optical densities were converted into hemoglobin concentration changes using the Modified Beer–Lambert Law (MBBL). Signals were then filtered using an FIR(71) band-pass filter at 10–400 mHz. Additionally, for simplicity, a vector auto-regressive model (VAR
$\left(1\right)$) was used that describes the time dependency and Granger causality between oxy-hemoglobin channels. The subject’s connectivity is a relation between the channels. The results are a low-resolution map of connections between different Brodmann regions, as shown in Table 3. If long-lag dependency models are used, we highly recommend using a partially directed coherence estimator on the estimator X to enable optimal visualization and straightforward interpretation. It should be noted that $\Phi_{0}^{\left(\ell \right)}$ models the connectivity for the baseline (barefoot), while $\Phi_{1}^{\left(\ell \right)}$ models the connectivity difference (in comparison to the baseline) when the subject wears a flat-sole sandal. Similarly, $\Phi_{2}^{\left(\ell \right)}$, $\Phi_{3}^{\left(\ell \right)}$, and $\Phi_{4}^{\left(\ell \right)}$ model the changes in connectivity that occur when the participant wears a medially wedged sandal, personal shoes, or SGL AS shoes (with wedges when necessary).

### 3.1. Low-Rank Representation

We use a low-rank projection operator Ψ to map the input *L* channels into *M* “informative” channels to provide a more interpretable result and feasible computational model. Connectivity models require high-quality recordings of all channels at all times [26], and a projection can mitigate motion artifacts [32]. The following approach is used to define the operator Ψ:A coordinate matrix $C\in R^{L\times 3}$ is constructed containing the 3D coordinates $\left\{c_{k}\right\}\;k=1,\ldots L$.Using a *K*-means method with random initialization and Euclidean distances, we find *M* clusters. As a result, each channel *k* is assigned a label ℓ=1…M. These clusters characterize spatially connected areas.The $g_{\ell }\left(t\right)$ is defined as the median value of $\left\{y_{c}\left(t\right)|\ell_{c}=\ell \right\}$ for each label *ℓ*.There is a strong rationale for not using an alternate estimator such as the mean $g_{\ell }^{*}\left(t\right)$:
(1)$$g_{\ell }^{*}\left(t\right)=\frac{1}{\sum_{c}I\left[\ell_{c}=\ell \right]}\sum_{c|\ell_{c}=\ell }y_{c}\left(t\right)$$The median $g_{\ell }\left(t\right)$ is a robust estimator that can reduce the impact of outliers in the estimations. Therefore, even when a few channels have a low signal-to-noise ratio, the median may provide a more accurate description of the hemodynamic signals in a particular region.Finally, let $\Psi Y\left(t\right)$ be the projected time series defined by
(2)$$\Psi Y\left(t\right)=G\left(t\right)=\left(\begin{array}{c} g_{1}\left(t\right)\\ g_{2}\left(t\right)\\ \vdots \\ g_{M}\left(t\right) \end{array}\right)\in R^{L\times M}$$

Although more robust and statistically sophisticated techniques are available, the above strategy achieves a reasonable trade-off between interpretability, complexity, and robustness.

### 3.2. Brain Dynamics: Condition-Driven Effective Connectivity

The projected time series $\Psi Y\left(t\right)$ is assumed to be weakly stationary and to admit a vector auto-regressive (VAR) representation of order *p*, also denoted as VAR(p) [33]:(3)$$\Psi Y\left(t\right)=\sum_{\ell =1}^{p}\Phi^{\left(\ell \right)}\Psi Y\left(t-\ell \right)+\varepsilon \left(t\right)$$
where is zero-mean multivariate white noise ″$\left(t\right)$ with covariance matrix $\Sigma_{\epsilon }$, and $\left\{\Phi_{1},\Phi_{2},\ldots ,\Phi_{L}\right\}$ are M×M coefficient matrices.

Despite their linear nature, VAR models can capture the time dynamics [34] and the spectral characteristics of a wide variety of systems [35,36].

We hypothesized that each condition of the Infinity Walk experiment (Table 2) would change brain connectivity. These complex condition-dependent signal dynamics can be modeled by assuming that the coefficients in (Equation 3) are affected by a set of K+1 dichotomic conditions:(4)$$\Phi^{\left(\ell \right)}=\Phi_{0}^{\left(\ell \right)}+\sum_{k=1}^{K}\Phi_{k}^{\left(\ell \right)}C_{k}$$

Then (Equation 3) and (Equation 4) jointly described the system dynamics by $\Phi_{k}^{\left(\ell \right)}$, which is allowed to change according to the experimental condition $C_{k}$. The *p*-values of estimators of $\hat{\Phi }$ can be evaluated due to the known limiting properties of the vectorized Φ, $\mathrm{vec}\left(\hat{\Phi }\right)$:(5)$$\mathrm{vec}\left(\hat{\Phi }\right)\to N\left(0,\Sigma_{\Phi }\right)$$

To understand the explainability of the model, let us consider a system with three channels, i.e., i,j,k, with $G\left(t\right)=\Psi Y\left(t\right)$ and p=1, as shown in (Equation 6): (6)$$\begin{array}{c} \begin{array}{c} g_{i}\left(t\right)=\underset{\begin{array}{c} \mathrm{Effect}\;\mathrm{of}\;i\\ \mathrm{at}\;t-1 \end{array}}{\underbrace{\phi_{i\to i}^{\left(1\right)}}}g_{i}\left(t-1\right)+\underset{\begin{array}{c} \mathrm{Effect}\;\mathrm{of}\;j\\ \mathrm{at}\;t-1 \end{array}}{\underbrace{\phi_{j\to i}^{\left(1\right)}}}g_{j}\left(t-1\right)\\ +\underset{\begin{array}{c} \mathrm{Effect}\;\mathrm{of}\;k\\ \mathrm{at}\;t-1 \end{array}}{\underbrace{\phi_{k\to i}^{\left(1\right)}}}g_{k}\left(t-1\right)+\underset{\begin{array}{c} \mathrm{Variations}\;\mathrm{not}\\ \;\left(\mathrm{explicitly}\right)\;\\ \;\mathrm{modeled}\; \end{array}}{\underbrace{\varepsilon \left(t\right)}} \end{array} \end{array}$$

Each $\phi_{j\to i}^{\left(1\right)}$ represents the impact of previous values of the channels into *j*. This impact can vary according to the experimental condition:(7)$$\begin{array}{c} \begin{array}{c} \phi_{j\to i}^{\left(1\right)}=\underset{\begin{array}{c} \mathrm{Effect}\;\mathrm{of}\;j\;\mathrm{on}\;i\\ \mathrm{baseline}\;\mathrm{when}\\ C_{1}=\cdots =C_{5}=0 \end{array}}{\underbrace{\phi_{0,j\to i}^{\left(1\right)}}}+\underset{\begin{array}{c} \mathrm{Effect}\;\mathrm{of}\;j\;\mathrm{on}\;i\\ \mathrm{due}\;\mathrm{to}\;\mathrm{condition}\;1 \end{array}}{\underbrace{\phi_{1,j\to i}^{\left(1\right)}}C_{1}}\cdots \\ +\underset{\begin{array}{c} \mathrm{Effect}\;\mathrm{of}\;j\;\mathrm{on}\;i\\ \mathrm{due}\;\mathrm{to}\;\mathrm{condition}\;5 \end{array}}{\underbrace{\phi_{5,j\to i}^{\left(1\right)}}C_{K}} \end{array} \end{array}$$

The coefficient ${\hat{\phi }}_{k,j\to i}^{\left(\ell \right)}$ is estimated using an ordinary least squares (OLS) method. Note that assuming a null hypothesis of $\phi_{k,j\to i}^{\left(1\right)}=0$, we can obtain a respective *p*-value that measures our confidence against the absence of connectivity between two (summarized) channels. Consequently, respective “t-value maps” or “*p*-value maps” can be constructed from these coefficient estimates. For simplicity in the estimation, this model is applied over the mean signal $E_{\tau }\left[\Psi Y_{\gamma }\left(t\right)\right]$ across trials γ∈Γ and with a first-order VAR model.

As a consequence of the proposed model, an ordinary least-square algorithm can be efficiently used to estimate its parameters with linear time complexity O(LM), which increases linearly with the number of channels *M* evaluated and the sample size *L*. In comparison, deep-learning connectivity methods using long short-term memory networks have a time complexity $O(R\left(C^{2}+CLM\right))$ for *R* training iterations and *C* memory cells [37].

Using the barefoot condition $C_{0}$ as a reference is an efficient approach for evaluating connectivity during a walking experiment using brain signals since it offers a baseline or control condition $\Phi_{0}$ against which to evaluate the other four scenarios of walking-induced brain networks. For completeness, recall that the conditions presented in Table 2 are encoded as follows:$$\begin{array}{cc} C_{0} & =\left\{\begin{array}{cc} 1 & \mathrm{Baseline}\;\left(\mathrm{barefoot}\right)\\ 0 & \mathrm{other} \end{array}\right.\\ C_{1} & =\left\{\begin{array}{cc} 1 & \mathrm{Flat}\;\mathrm{sole}\\ 0 & \mathrm{other} \end{array}\right.\\ C_{2} & =\left\{\begin{array}{cc} 1 & \mathrm{Flat}\;\mathrm{sole}\;\mathrm{with}\;\mathrm{medial}\;\mathrm{heel}\;\mathrm{wedge}\\ 0 & \mathrm{other} \end{array}\right.\\ C_{3} & =\left\{\begin{array}{cc} 1 & \mathrm{Personal}\;\mathrm{shoes}\\ 0 & \mathrm{other} \end{array}\right.\\ C_{4} & =\left\{\begin{array}{cc} 1 & \mathrm{SGL}\;\mathrm{Technology}/\mathrm{GAITLINE}\;\mathrm{shoes}\\ 0 & \mathrm{other} \end{array}\right. \end{array}$$

## 4. Results and Discussion

The results obtained in Figure 4 depict 2D topographic plots of statistically significant connections (*p*-value ≤ 0.05, $P(\phi_{\ell ,i,j}\neq 0)$ in Equation (Equation 3)) in each participant’s estimated connectivity maps $\hat{\Phi }$. The results are the preliminary understanding of the effective connectivity (hereby only called connectivity) between different footwear conditions while performing the Infinity Walk. Overall, significantly low connectivity was observed between different brain ROI (motor, somatosensory, and temporal areas) in barefooted conditions except in Subject 2. The low connectivity in barefoot walking is due to it being the most realistic walking condition, requiring optimal motor control, attention, and somatosensory demands [22]. On the contrary, the highest interconnection between the ROI was observed when the participants performed the Infinity Walk with a flat insole and medially wedged sandals. Comparatively, walking with a flat-insole sandal with or without medial wedges was challenging for the participants during walking. The condition is challenging as all subjects have several degrees of pronation (see Table 1) as a natural condition—with a flat sole, it was hard to perform the Infinity Walk while maintaining postural balance. Using the medial wedges under the heel area, we initiated a change in the motoric pattern by placing the heel bone in a vertical position. When the heel bone rotates medially (heel valgus), less skin contacts the ground; thus, less lateral mechanosensory stimulation is expected. When the heel skin is laterally pulled, it provokes a sensation of balance instability. It may feel like a new sensation to the subjects due to heel valgus and a different sense of balance. Aligning the heel bone in a vertical position is “a correction to increase the balance-readiness” (IBR). A vertical heel bone probably activates balance responses and postural stability responses (for instance, along the spine), which we believe is a positive postural response for most people for the caretaking of lumbar spine control. Another reason could be that walking with flat sandals was performed right after a barefoot walk. Further, the differences in the outsole and upper construction of the sandals and shoe (own shoe or SGL technology shoe) have a different level of feeling and confidence in walking, which might affect the motor control and hence the results. Therefore, in future studies, instead of sandal shoes with flat insoles or wedges, a design similar to shoes (own shoes or SGL technology shoes) is needed to make the overall construction of the footwear consistent.

The results demonstrate no significant difference in connectivity patterns (flat sole and medially wedged sandals) corresponding to overpronation. In the case of Subjects 4 and 5 with corrected medially wedge sandals, the connectivity patterns decreased compared to those of flat-insole sandals. However, the effect of overpronation on connectivity is hard to conclude because of the lack of control among participants and the experimental design. The highest connectivity was found using flat-sole sandals across all the subjects, but exclusive remarks cannot be concluded to show the effect of overpronation on connectivity. In the current setting, there could be multiple reasons for higher connectivity among the brain region; among them could be difficulty walking and maintaining balance. Considering lateralization of the brain, the participant with more medial wedge correction on the right side than on the left side has more connectivity networks between the ROI in the right hemisphere. Another reason for higher connectivity wearing flat and medially wedged sandals might be slippage of the wedges, which makes walking challenging with the Infinity Walk, as shown in Figure 1. We used a strong and thin adhesive double-sided tap to reduce the slippage. It helped to reduce the slippage but added more attentional bias to it. A more sophisticated medially wedged insole design is needed in the future to reduce the difficulty of walking. Strong connectivity between the temporal and motoric regions was found in the right hemisphere compared to the left side, particularly in challenging conditions such as flat-insole sandals and medially wedged sandals. A recent study shows that the temporal region is not just a speech area; it would be interesting to investigate it from a balanced perspective in the future [38].

The SGL technology shoe is featured to reduce overpronation. The most important part of the construction is the heel-wedge, which is approximately 7 degrees, since it is in the heel area where the trace of the CoP (from heel to toe) starts. With heel valgus, the trace goes medially in the foot. With heel varus, the trace goes laterally. The second part is the arch roller in combination with the shank, which supports the foot arch (the plantar aponeurosis). Since all materials in the outsole construction are flexible and dynamic, the total impact on the foot is a dynamic, progressive stimulus. It is progressive as a greater tendency to overpronate causes more corrective support from the technical components in the outsole. No significant difference was found in the connectivity pattern other than compared to the barefooted condition. The reasons include new postural mechanisms and feeling and emotional involvement of wearing a new shoe (SGL Technology). The evidence is that comparatively fewer connections between ROI were found when the subjects were wearing their own shoes. The participant were given an SGL shoe, which will be tested in the future after a couple of months to support the argument further. After walking in the SGL Technology shoes, people will get used to the new heel bone and mid-foot positions, and brain activity might be reduced.

### 4.1. Infinity Pattern

An interesting fact is that due to the Infinity Walk, there was a bidirectional connective relation between both hemispheres and ROI across all the conditions. A demonstration of association connectivity during the Infinity Walk and Infinity Pattern is shown in Figure 5. It would be interesting to see the changes with time perspective in the future corresponding to cortical activation. This verifies that the Infinity Walk can be a powerful tool for motoric learning, neuro-rehabilitation, and balanced walking patterns [8,9]. The Infinity Walk is homogeneous to juggling for lower extremities because of its progression of sensorimotor, cognitive language, and other multi-tasking skills while maintaining attention and balance. Our results verify the interconnection of multiple ROI during the Infinity Walk, which could be a good tool for regenerating the neural networks in cases of motor disabilities.

### 4.2. Limitations and Future Directions

The number of participants and trials in the study is limited, and we cannot generalize the results to a larger population and older ages. However, the study laid the foundation for further experimentation. The rest time between two consecutive movements was 10 seconds, which was not enough time to settle the hemodynamics back to baseline. It needs to be increased in future experimentation. It should be noted that a low-rank representation is not necessarily required for the connectivity operations mentioned in Section 3.1. Nonetheless, another strategy should be explored without such a mechanism in order to eliminate the influence of defective channels in later analyses. In the future, more sophisticated signal pre-processing methods need to apply before connectivity analysis to reduce the physiological noise. The effect of correct medial wedge angle (posture correction) on cortical connectivity needs further investigation because with the current setting it was difficult to conclude a relationship between both. In the future, a simple straight walk should be performed to separately investigate the pronation effect on effective connectivity between brain regions. Additionally, to further strengthen the results of the Infinity Walk, the Infinity Walk must be compared with a simple straight walk. Therefore, the experiment’s findings could describe the underlying neural mechanisms that drive the Infinity Walk realistically.

## 5. Conclusions

In conclusion, connectivity between the regions and hemispheres increases as the difficulty of walking increases. More and more regions in the motoric, sensorimotor, and temporal regions were connected during complex and difficult footwear and walking patterns. Barefoot walking has less connectivity compared to other conditions. The Infinity Walk seems promising, and it can be effectively used to increase motor learning because it constantly engages and causes communication between regions in both hemispheres.

## Figures and Tables

**Figure 1 sensors-23-04422-f001:**
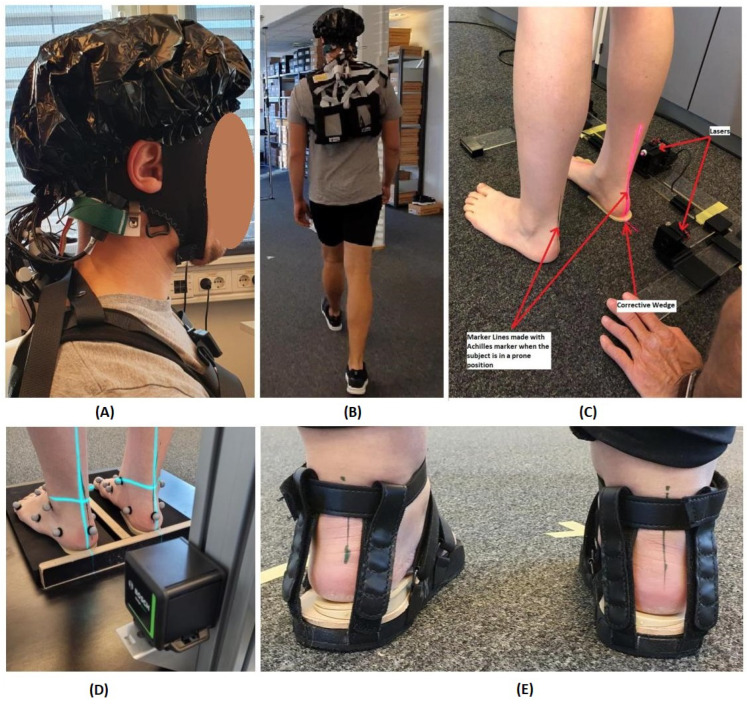
Experimental setup: (**A**) fNIRS optodes covered with a shower cap to reduce environmental light; (**B**) demonstration of the subjects performing Infinity Walk with fNIRS equipment; (**C**) Achilles laser correction line setup; (**D**) medial wedge correction in posture correction equipment; (**E**) flat-insole sandal with additional medial wedges to correct the posture.

**Figure 2 sensors-23-04422-f002:**
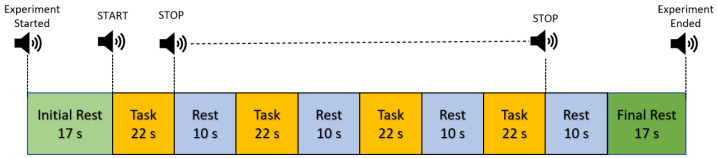
Experimental protocol: single trial consisted of task (22 s) followed by rest (10 s). Four repetitive trials were conducted for each condition mentioned in Table 2.

**Figure 3 sensors-23-04422-f003:**
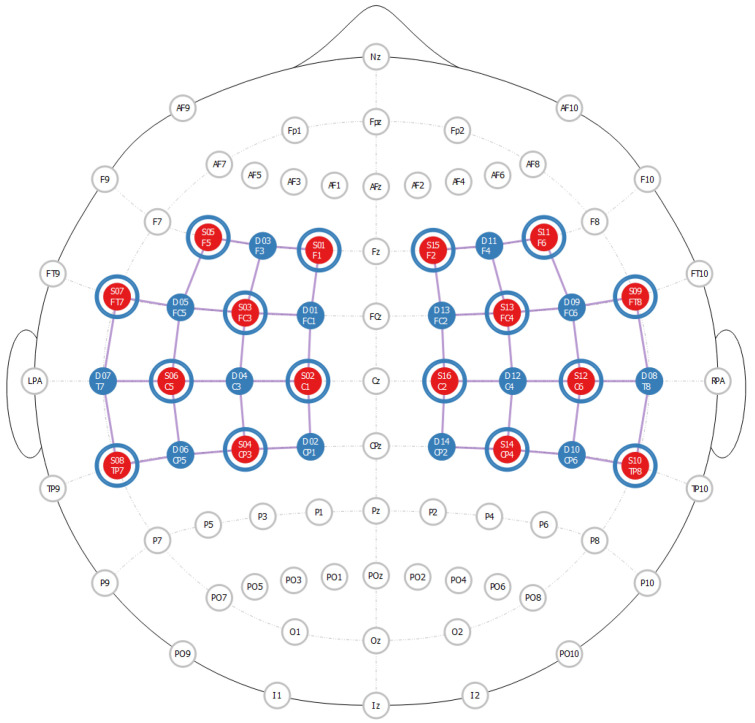
Optode configuration over the regions of interest.

**Figure 4 sensors-23-04422-f004:**
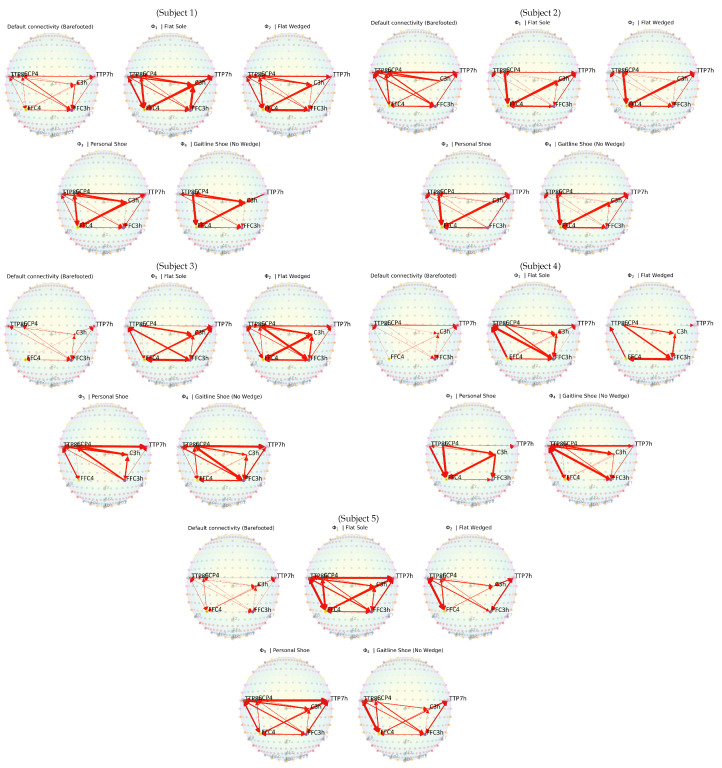
Connectivity maps obtained from the five subjects involved in this experiment according to the type of condition.

**Figure 5 sensors-23-04422-f005:**
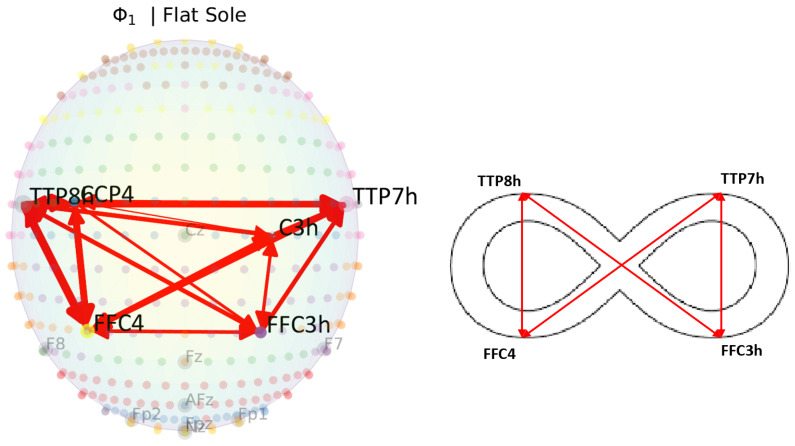
Association of Infinity Walk to Infinity Pattern of connectivity.

**Table 1 sensors-23-04422-t001:** The medial heel wedges on both sides of each subject.

Subjects	Left Leg (Degrees)	Right Leg (Degrees)
01	12	06
02	08	11
03	17	17
04	12	09
05	17	17

An additional 5 mm flat heel lift was added to the subject’s right leg to balance the height of the legs.

**Table 2 sensors-23-04422-t002:** List of conditions tested during Infinity Walk.

Condition	Description
1	Barefooted walk on plain surface
2	Walking with flat-insole sandals
3	Walking with pronation-corrected sandals or medially wedged sandals
4	Walking with personal shoes
5	Walking with SGL technology shoe, referred to as GAITLINE AS shoes

**Table 3 sensors-23-04422-t003:** Brain regions of interest (ROI).

Sr. No.	Brain Region	Shared Brodmann Areas	EEG 128 Positions	Description
1	Motoric areas	6, 8, 44	FFC4h, FFC3h	Agranular frontal, Intermediate frontal, Opercular
2	Somatosensory area	7, 40	CCP4, C3h	Superior parietal, Supramarginal
3	Temporal area	21, 22, 43	TTP8h, TTP7h	Middle temporal, Superior temporal, Subcentral

## Data Availability

Data set is available upon request.

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
