# Peer review of "Brain Connectivity Analysis in Distinct Footwear Conditions during Infinity Walk Using fNIRS"

_sensors, 2023, doi:10.3390/s23094422_

Round 1
Reviewer 1 Report
Summary
This work examines the brain connectivity between different brain regions during the infinity walk with different footwear conditions/pronation. Granger’s causal connectivity analysis was performed. This exploratory initial study is interesting and the authors are also encouraged to perform more robust studies in the future. The experimental protocol was simple and straightforward and limited to 5 subjects (not conclusive). This is a starting point for the next project. In my opinion, this work is publishable in MDPI Sensors with a few minor modifications.
Abstract/Introduction
· Line # 12-13. If authors could put some numerical results. It would improve the abstract.
· Reference to the Granger causal model in line # 90 is needed. “Seth, A.K.; Chorley, P.; Barnett, L.C. Granger causality analysis of fMRI BOLD signals is invariant to hemodynamic convolution but not downsampling. NeuroImage 2013, 65, 540–555. https://doi.org/10.1016/j.neuroimage.2012.09.049”
· If any toolbox is used, it should be mentioned later in the paper. “The MVGC toolbox is currently under development. It is intended to supersede the Granger causal connectivity analysis’ (GCCA) toolbox (Seth, 2010)”.
· Line # 69, citations of laboratory-made fNIRS systems by other researchers are needed.
· Line# 71. Also, cite other applications of the fNIRS system such as cognitive load published in MDPI Sensor https://www.mdpi.com/1424-8220/21/11/3810
Materials
· How the author addressed the issue with the slow walker and fast walker within the 22 seconds? If 22 seconds is the limit, how many gait cycles per participant (or mean standard deviation) may be included in the paper? (Line #134)
· Are the four trials randomized or sequential?
· Lines # 130 to 138 are confusing. Please clarify/rewrite the sentences so that readers can easily understand. Table 2 has 5 conditions. Each of the four tasks in the timing block diagram (you are referring to as protocol) are of with one condition (from Table 2)
Connectivity Model
· Line # 194. Spacing.
· Line # 151. What is M
· Line # 170. What is the value of M "informative" channels obtained.
· The equations are generic. From the connectivity model, what model parameters obtained/set should be mentioned in the last paragraph.
Results
· Line # 218. 2-D topographic plot.
· In the limitation section (Line #295). Please include that number of trials is limited in this study. Also, sophisticated signal pre-processing methods are not used before connectivity analysis. This could be in your future studies.

Author Response
Dear Reviewer,
We appreciate your time and effort devoted to providing feedback on our manuscript. We are grateful for the insightful comments and valuable improvements to our paper. Most of the suggestions are incorporated in the revised version. The point-by-point response is attached in a pdf file.
Best regards,
Haroon Khan et al.

Reviewer 2 Report
Authors examines the effect of infinity walk, pronation, and footwear condition on brain-effective connectivity patterns. This study is valuable and article is well written. I recommend authors to look following study to improve the discussion
Please see “Towards the Development of Versatile Brain-Computer Interfaces”. If it looks difficult for you to perform experiments with large number of subjects at least discussed the above-mentioned study.
2. The classification results for cross subject motor imagery are very low for the proposed method. Have a look on following study where authors achieved 98.3% classification accuracies “Motor Imagery BCI Classification Based on Multivariate Variational Mode Decomposition”.
3. For EEG Classification, first step is denoising of signals, MSPCA plays vital role, which is a combination of PCA and wavelet. I recommend authors to use MSPCA in discussion.
4. For EEG Classification, convolutional neural networks play vital role, thus details should be included in literature.
5. Graphical features are one of the newest approaches for identifying underlying patterns of EEG signals and details of these methods can significantly increase the readability.
6. Please provide the details of future direction and possible solutions to continue this topic.
Finally, I suggest authors to sit with English native speaker to improve the writing of proposed work
Author Response

(The authors gave the same response as above.)

Round 2
Reviewer 2 Report
looks good now